# Osteogenic Protection against Fine Dust with Erucic Acid-Induced Exosomes

**DOI:** 10.3390/jfb15080215

**Published:** 2024-07-31

**Authors:** Hyunjung Kim, Boyong Kim

**Affiliations:** 1Department of Health and Safety Convergence Science, Korea University, Anam-ro 145, Seongbuk-gu, Seoul 02841, Republic of Korea; hjkim99@korea.ac.kr; 2EVERBIO, 131, Jukhyeon-gil, Gwanghyewon-myeon, Jincheon-gun 27809, Republic of Korea

**Keywords:** adipose-derived stem cells, macrophage, erucic acid, PM10, microvesicle

## Abstract

Fine dust causes various disorders, including cardiovascular, neurological, renal, reproductive, motor, systemic, respiratory, and cancerous diseases. Therefore, it is essential to study functional materials to prevent these issues. This study investigated the beneficial effects of erucic acid against fine dust using methods such as miRNA profiling, quantitative PCR, flow cytometry, ELISA, and Alizarin O staining. Erucic acid effectively suppresses inflammation and upregulates osteogenic activators in fibroblasts exposed to fine dust. Additionally, erucic acid-induced exosomes (EIEs) strongly counteract the negative effects of fine dust on osteocytic differentiation and inflammation. Despite fine dust exposure, EIEs promoted osteocytic differentiation in adipose-derived stem cells (ASCs) and enhanced osteogenesis and phagocytosis in macrophages. The significant upregulation of RunX2 and BMP7 by EIEs indicates its strong role in osteocytic differentiation and protection against the effects of fine dust. EIEs also boosts immune activity and acts as an osteogenic trigger for macrophages. MicroRNA profiling revealed that EIEs dramatically upregulated miRNAs, including hsa-miRNA-1301-3p, hsa-miRNA-1908-5p, hsa-miRNA-423-5p, and hsa-miRNA-122-5p, which are associated with osteogenic differentiation and immunity. Therefore, EIEs show potential as biomaterials to prevent environment-borne diseases.

## 1. Introduction

Fine dust, or particulate matter, can be found in the air, and fine dust particles sized 2.5 μm (PM2.5) and 10 μm (PM10) are very harmful to the human body [1]. In particular, elderly people and children are more susceptible to fine dust [2,3]. Fine dust has been shown to cause various disorders, including cardiovascular, neurological, renal, reproductive, motor, systemic, respiratory, and cancerous diseases [4,5]. According to a recent study [6], fine dust suppresses the osteogenic differentiation of adipose-derived stem cells. Moreover, fine dust causes inflammation in the dermal tissues in humans and pets [6]. Unlike PM2.5, which is at a low concentration in summer and high in winter, PM10 is high in spring and winter [7]. This high exposure rate to PM10 results in worse conditions skin. Skin inflammation causes adverse dermal immunity leading to differentiation of adipose-derived stem cells (ASC_S_) in the subcutaneous fatty tissues of the skin [6,8]. When exposed to fine dust, dermal cells upregulate apoptotic proteins, including BAX and CytC, and downregulate antiapoptotic proteins, including AKT, P50, P52, and BCL-2 [6]. Notably, fine dust enhances differentiation of osteoclasts from stem cells [9].

Furthermore, according to recent reports, exposure to fine dust is one reason for osteoporosis and increases the risk of contracting the disease [10].

Osteogenesis is important for bone regeneration associated with age-related bone diseases [11]. Changes in the expression of various genes occur during the osteogenic stages of mesenchymal differentiation. The levels of RUNX2 and DLX5 are increased in preosteoblasts, while osteoponin and SOST-sclerostin are expressed in osteoblasts and osteocytes, respectively [6,12,13]. In contrast, p53 downregulates RUNX2 and osterix in ASC_S_ [13]. Ordinarily, inflammation inhibits osteogenic differentiation [14]. Expression of VEGF and TGF-β, known as osteogenic activating markers, are downregulated in fibroblasts during inflammation. Additionally, aromatase, collagenase type 1, RUNX2, SAMD4, BMP6, and BMP7 are known activators of the osteogenic differentiation of ASC_S_ [15,16]. In the immune system of bone homeostasis, macrophages secrete cytokines, including TNF-α, IL-1, and IL-6, to inhibit bone formation and the cytokines, such as IL-4, IL-10, and IL-13, inhibit the differentiation of osteoclasts [17].

Erucic acid, a monounsaturated omega-9 fatty acid, is found in seed oils of various plants, especially from the *Brassicaceae* family [18,19]. According to a previous study [20], rapeseed, mustard, and wallflower seed oils contain high concentrations of erucic acid. This compound is used in the treatment of toxic oil syndrome and reduces cardiotoxicity in rats [21,22]. The beneficial effects of erucic acid include therapeutic effects in neurodegenerative diseases [19], obesity-induced metabolic disorders, melanomas, and diabetes. It also has anti-inflammatory and antioxidant properties [23]. Although erucic acid is classified as a toxicant, the maximum edible quantity for this compound is 7.5 mg/kg body weight/day [24].

Exosomes are approximately 40–100 nm in size, and almost all cell types secrete them into either the serum, urine, cerebrospinal fluid, ascites fluid, milk, or saliva [25,26]. Exosomes contain various molecules, including functional protein, carbohydrate, mRNA, microRNA (miRNA), and DNA molecules [27]. In general, cells exposed to a stimulant, upon induction, secrete exosomes that dramatically alter their components compared to unstimulated conditions [28]. Altered exosomes can have adverse or beneficial effects on the surrounding cells [28]. In addition, exosomes have various functions, including the modulation of the immune system, prognostic biomarkers for diseases, and cancerous activity in the human body [26,28]. Due to these characteristics, induced exosomes have the potential to be used as biofunctional materials in various fields, including pharmaceuticals, cosmetics, and foods [28].

Various factors [29,30] and the deterioration of the environment have worsened the problem of fine dust [6]. Additionally, chemical compounds are used in various products and play a crucial role, particularly in cosmetics, pharmaceuticals, and functional foods [31]. However, their side effects must be seriously considered by industries [31]. To overcome these issues, the development of biomaterials has become increasingly necessary. Therefore, it is necessary to study functional biomaterials with minimized side effects to prevent various dermal problems arising as a consequence of fine dust. Nevertheless, there is limited research on the osteogenic effects of fine dust on dermal and immune cells.

The goals of this study are to describe the suppression of osteogenic differentiation by fine dust in inflammatory fibroblasts and to propose a new biomaterial that prevents bone diseases and immune problems caused by fine dust.

## 2. Materials and Methods

### 2.1. Cell Culture

Fibroblasts (Korea Cell Bank, Seoul, Republic of Korea) were cultured in Dulbecco’s modified Eagle’s medium (DMEM) (Gibco, Thermo Fisher Scientific, Waltham, MA, USA) supplemented with 10% heat-inactivated fetal bovine serum (FBS) (Sigma-Aldrich, St. Louis, MO, USA), 100 µg/mL penicillin (Gibco, Thermo Fisher Scientific), and 100 µg/mL streptomycin (Gibco, Thermo Fisher Scientific) at 37 °C and 5% CO_2_. Adipose-derived stem cells (ASC_S_) (Thermo Fisher Scientific, Waltham, MA, USA) were cultured in a MesenPRO RS™ Basal Medium (Gibco, Thermo Fisher Scientific) with growth supplement (MesenPRO RS™ Growth Supplement, Thermo Fisher Scientific). Macrophages (KG-1, ATCC) were cultured in Iscove’s modified Dulbecco’s medium (IMDM) supplemented with 10% heat-inactivated fetal bovine serum (FBS) (Gibco, Thermo Fisher Scientific), 100 µg/mL penicillin (Gibco, Thermo Fisher Scientific), and 100 µg/mL streptomycin (Gibco, Thermo Fisher Scientific) at 37 °C and 5% CO_2_.

### 2.2. Cell Viability Test 

To establish treatment dosages of erucic acid and fine dust, fibroblasts were exposed to 0, 5, 50, 100, 500, and 1000 µM of erucic acid (45629, Sigma-Aldrich) and 0, 1, 5, 10, 20, and 50 µg/mL of PM10 (ERMCZ120; Sigma-Aldrich) for 1 d, respectively. For the induced exosomes (S1), ASC_S_ and macrophages were exposed to 0, 1, 5, 10, 50, or 100 ng/mL of induced exosomes to form the four conditions (control, erucic acid, fine dust, and erucic acid + fine dust) for 1 d. To evaluate viability, all exposed cells were stained with Annexin V-conjugated propidium iodide (PI) (Invitrogen, Carlsbad, CA, USA) and analyzed using a flow cytometer (FACS calibur, BD Biosciences, San Jose, CA, USA) with the FlowJo 10.10 software (BD Biosciences).

### 2.3. Evaluation of the Concentration of Induced Exosomes

After exposure to each of the four conditions (control, erucic acid, fine dust, and erucic acid + fine dust) for one day, the supernatants from the exposed fibroblasts were collected and then the induced exosomes (CE, EIE, FDIE, and EFDIE) were isolated and purified from the supernatants (10 mL) using the exoEasy Maxi Kit (QIAGEN, Hilden, Germany) and CD68 Exo-Flow Capture Kit (System Biosciences, Palo Alto, CA, USA), respectively, and the concentrations of the isolated exosomes were evaluated using the exosome standards kit (Sigma-Aldrich). To avoid contamination from other extracellular vesicles, fibroblasts were cultured without fetal bovine serum. Purified exosomes were evaluated using a flow cytometer (FACSCalibur, BD Biosciences) and FlowJo 10.10 software (BD Biosciences). 

### 2.4. Quantitative PCR 

Total RNA was extracted from cells (fibroblasts and ASC_S_) using the RiboEx reagent (GeneAll, Seoul, Republic of Korea). In reverse transcription, the extracted RNA was then reverse-transcribed into cDNA using a Maxime RT PreMix (iNtRON, Seongnam, Republic of Korea). For quantitative PCR, the synthesized cDNAs were amplified using specific primers (Table 1) with the following cycling parameters: 1 min at 95 °C, followed by 35 cycles of 35 s at 59 °C and 35 cycles of 1 min at 72 °C. The expression levels of the target genes in the samples were normalized to those of the housekeeping gene GAPDH, and the relative quantities of the target genes were determined with respect to those of the control.

### 2.5. Phagocytic Activity Test 

The cultured macrophages were exposed to the four types of induced exosomes (CE, EIE, FDIE, and EFDIE) for one day, and the exposed cells were treated with FITC-labeled E. coli particles for 3 h using a Phagocytosis Assay Kit (ab235900, Abcam, Cambridge, UK). Negative controls were acquired from cultured macrophages without induced exosomes. The treated cells were analyzed using a flow cytometer (BD FACSCalibur) with the FlowJo 10.7.0 software (BD Biosciences).

### 2.6. Evaluating of Cytokine Concentration

After the cultured macrophages were exposed to the four types of induced exosomes (CE, EIE, FDIE, and EFDIE) for one day, their culture media were isolated. The isolated media (1 mL) were centrifuged at 1000× *g* for 10 min to remove debris, and cytokines in the isolated media were evaluated with standard samples (1, 2.1, 4.4, 8.75, 17.5, 35 pg/mL) using IL-4 and IL-6 ELISA kits (ab46058 and ab178013, Abcam) and a microplate reader (AMR-100; Allsheng, Hangzhou, China).

### 2.7. Alizarin O Staining 

After the cultured macrophages were exposed to the four types of induced exosomes (CE, EIE, FDIE, and EFDIE) for 5 and 15 days, the cultured cells were fixed with 2% paraformaldehyde for 12 h. For staining, the exposed cells were treated with Alizarin O reagent (Sigma-Aldrich) for 40 min. The stained cells were evaluated using a fluorescence microscope (Eclipse Ts-2; Nikon, Shinagawa, Japan) and imaging and counting software (NIS-elements V5.11 (Nikon)).

### 2.8. Profiling of microRNA in MSEIEs

The isolated and purified exosomes were sequenced by ebiogen Inc. (Seoul, Republic of Korea) to analyze exosomal functions. An Agilent 2100 bio-analyzer and the RNA 6000PicoChip (Agilent Technologies, Amstelveen, The Netherlands) were used to evaluate RNA quality. RNA was quantified using a NanoDrop 2000 spectrophotometer (Thermo Fisher Scientific, Waltham, MA, USA). Small RNA libraries were prepared and sequenced using the Agilent 2100 Bio-analyzer instrument for the high-sensitivity DNA assay (Agilent Technologies, Inc. Santa Clara, CA, USA) and NextSeq500system single-end 75 sequencing (Illumina, San Diego, CA, USA). To obtain an alignment file, the sequences were mapped using bowtie 2 software (CGE Risk, Lange Vijverberg, the Netherlands), and the read counts were extracted from the alignment file using bedtools (v2.25.0) (GitHub, Inc., San Francisco, CA, USA) and R language (version 3.2.2) (R studio, Boston, MA, USA) to evaluate the miRNA expression level. miRWalk 2.0 (Ruprecht-Karls-Universität Heidelberg, Medizinische Fakultät Mannheim, Germany) was used for miRNA target signal study, and ExDEGA v.2.0 (ebiogen Inc., Seoul, Republic of Korea) was used to deduce radar charts.

### 2.9. Statistical Analysis 

All experiments were analyzed by one-way analysis of variance (ANOVA) with the post hoc test (Scheffe’s method) using Prism 7 software (GraphPad, San Diego, CA, USA). *p* values (*p* < 0.05, <0.01 and <0.001) were statistically significant.

## 3. Results

We studied the biological functions of erucic acid and erucic acid-induced exosomes in the prevention of osteogenic suppression and inflammation caused by fine dust exposure to dermal cells, immune cells, and ASC_S_.

### 3.1. Antiapoptotic Function of Erucic Acid in Fibroblasts Exposed to Fine Dust

Based on the obtained results (Figure 1), treatment dosages for erucic acid and fine dust (PM10) were established as 250 μmole/mL and 20 μg/mL, respectively, in fibroblasts (Figure 1a,b). Under the four conditions (CE, EIE, FDIE, EFDIE), the concentrations of exosomes isolated from fibroblasts were approximately 2 × 10^9^, 2.5 × 10^9^, 1.5 × 10^9^ and 2.8 × 10^9^ particles/mL, respectively (Figure 1c,d). The levels of osteogenic activators were increased upon exposure to erucic acid in fibroblasts (Figure 2). Upon erucic acid exposure, the VEGF levels were approximately six times higher, while the levels of TGF-β were four times higher, than upon FD exposure (Figure 2a,b). Although exposure to fine dust downregulated VEGF and TGF-β in fibroblasts, erucic acid prevented the suppression of the two markers in the cells (Figure 2a). Additionally, erucic acid enhanced the expression of the markers by approximately 3.2 times in comparison with the control (Figure 2a). Further, erucic acid induced the upregulation of antiapoptotic markers in fibroblasts in addition to protecting against fine dust (Figure 2b).

### 3.2. Protecting and Enhancing of Osteocytic Differentiation by the Induced Exosomes

The erucic acid-induced exosomes (EIE), isolated from fibroblasts, enhanced expression of the osteocytic differentiation markers, including aromatase, CT1 (collagenase type 1), RunX2, TGF-β, VEGF, SAMD4, BMP6, and BMP7 (Figure 3). In particular, RunX2, BMP 6, and BMP7 were dramatically upregulated in ASC_S_ exposed to EIEs. Despite exposure to fine dust, increased levels of the markers were observed in erucic-acid induced exosomes (EFDIE) (Figure 3a–c). The Alizarin staining results (Figure 4) corresponded with the results shown in Figure 3; ASC_S_ were strongly differentiated into osteocytes at day 5, and then formed osteocytic colonies under EIE and EFDIE conditions on day 15 (Figure 4).

### 3.3. Activation of Immunity in Macrophages with Induced Exosomes

Macrophages exposed to fine dust showed attenuated phagocytic activity; in contrast, this activity was strongly enhanced under EIE and EFDIE conditions (Figure 5a,b). Compared with the FD condition, the phagocytic activity was approximately 10 and 7.5 times higher under EIE and EFIE conditions, respectively (Figure 5a,b). Upon evaluation of osteogenesis-modulating cytokines (Figure 6), we found that osteogenesis-inhibiting cytokines (TNF-α, IL-6) were downregulated in macrophages under EIE and EFDIE conditions, in contrast to the FDIE conditions (Figure 6a,b). However, osteogenesis-activating cytokines were upregulated in macrophages under EIE and EFDIE conditions (Figure 6a,b). Notably, EIE significantly modulated IL-6 and IL-4 levels in macrophages (Figure 6b).

### 3.4. Profiling of miRNAs in Varios Exosomes

Under erucic acid, fibroblasts synthesized and secreted their induced exosomes (EIEs), which contained dramatic alterations in specific miRNAs (Figure 7). Based on the results of miRNA profiling, EIEs revealed dramatic alterations in miRNAs associated with modulation of four categories: activation of cellular differentiation, anti-apoptosis, anti-inflammation, and activation of DNA repair (Figure 7). Compared with FDIE, EIEs contained upregulated miRNAs, including has-miRNA-1301-3p, has-miRNA-1908-5p, has-miRNA-423-5p, and has-miRNA-122-5p (Figure 7 and Table 2). Notably, has-miRNA-423-5p was identified as a key candidate molecule for the four biological categories (Table 2).

## 4. Discussion

In this study, we divide functions of erucic acid and erucic acid-induced exosomes (EIE) into three categories: the activation of osteocytic differentiation, activation of osteogenesis by macrophages, and activation of the immune response.

First, erucic acid prevented fibroblasts from being inflamed upon exposure to 20 ng/mL fine dust (Figure 2). Recently, fine dust has been found to contribute significantly to respiratory, circulatory, and skin diseases [32,33]. Although fine dust causes an increase in ROS [34,35], erucic acid suppresses the stress in fibroblasts (Figure 2). Profiling of miRNAs in EIEs revealed upregulation of anti-apoptotic and inflammatory miRNAs (Table 2). The significant increase in hsa-miRNA-1908-5p and hsa-miRNA-423-5p corresponded to the anti-apoptotic function of erucic acid in fibroblasts (Figure 2 and Table 2). EIEs from fibroblasts exposed to erucic acid affected the anti-apoptotic response of fibroblasts under fine dust conditions.

Second, two materials, erucic acid and erucic acid-induced exosomes (EIEs) from fibroblasts, protected and activated osteocytic differentiation against fine dust. Erucic acid upregulated the osteogenic activators (Figure 2a). Additionally, EIEs increased the expression of the activating factors of osteocytic differentiation in adipose-derived stem cells (Figure 3a–c). VEGF, TGF-β, BMP6, and BMP7 were reported as a triggers for osteocytic differentiation in stem cells [36,37]. Additionally, in the pre-osteoblast stage, CT1 and RunX2 are upregulated during stem cell osteocytic differentiation [38,39]. In particular, RunX2 acts as an activator of bone matrix protein synthesis in immature osteoblasts [39]. As observed in Figure 3, EIEs intensely enhanced the expression of RunX2 and BMP 6, and BMP7. Based on the formation of Alizarin O staining (Figure 4), we can conclude that EIEs activate mineralization in differentiating ASC_S_. RunX2 plays the role of an activator of mineralization in immature osteoblasts to develop them to mature osteoblasts and osteocytes [35]. A dramatic increase in the number of calcium granules (Figure 4) was induced upon increase in the RunX2 levels during osteogenesis. Just as hsa-miRNA-1301-3p promotes osteogenic differentiation from stem cells [38], the upregulation of this miRNA, along with the other significant miRNAs in EIEs (Table 2), strongly activates osteogenic differentiation from ASCs (Figure 3 and Figure 4). These results suggest that EIEs trigger osteocytic differentiation and are a developmental activator of osteogenesis in stem cells. Moreover, EIEs play an effective protective role against fine dust in fibroblasts.

Third, EIEs activate modulation of osteogenesis by macrophages despite exposure to fine dust. Macrophages act as osteogenesis modulators and immune conductors in humans [40,41]. According to reports [42,43], TNF-α inhibits the clearance of apoptotic cells by macrophages, while IL-10 and IL-4 activate phagocytosis by macrophages. In the results shown in Figure 6, EIEs significantly upregulated IL-13 and IL-4, which are anti-inflammatory cytokines in macrophages. Interestingly, despite TNF-α typically activating macrophage phagocytosis, EIEs attenuated these levels in macrophages. These results suggest that the elevated levels of IL-13 and IL-4 played a crucial role in activating phagocytosis in macrophages (Figure 5 and Figure 6). Furthermore, the activities in EFDIE-exposed cells were stronger than those in FDIE-exposed cells, and their strength was approximately 7.5 times higher than that of FDIE-exposed cells (Figure 5). Ordinarily, hsa-miRNA-1908-5p and hsa-miRNA-423-5p modulate the immune response. Macrophages secrete cytokines, such as TNF-α, IL-1, and IL-6, which are associated with inhibiting bone formation and promoting bone resorption [17]. Additionally, IL-4, IL-10, and IL-13 secreted by macrophages inhibit osteoclast differentiation and OSM, while VEGF, IGF, TGF-β, and BMP-2 activate osteoblast differentiation [17]. In the results (Figure 6a,b), TNF-α and IL-6 were affected the most strongly by EIEs in the differentiating cells. These results suggest that EIEs prevent upregulation of TNF-α and IL-6 by fine dust and activate osteogenesis through upregulation of IL-4 and IL-13, despite exposure to fine dust, in differentiating cells. The ordinary phagocytic activity of immune cells is attenuated by fine dust [44]. EIEs significantly upregulated these miRNAs. These results suggest that EIEs are a very effective biomaterial for preventing the effects of fine dust and activating immunity and osteogenic differentiation.

## 5. Conclusions

This study comprehensively evaluates the multifaceted roles of erucic acid and erucic acid-induced exosomes (EIEs) across three critical biological processes: osteocytic differentiation, osteogenesis modulation by macrophages, and immune response activation. In summary, both erucic acid and EIEs demonstrate substantial potential as protective agents against the inflammation and oxidative stress induced by fine dust. Their ability to promote osteocytic differentiation and modulate macrophage activity highlights their effectiveness as biomaterials for enhancing immune responses and supporting bone health. These findings underscore the promising therapeutic applications of erucic acid and EIEs in mitigating the adverse effects of environmental pollutants and advancing regenerative medicine. Nevertheless, to fully realize their potential and apply them across various fields, it is essential to investigate the in vivo effects of EIEs corresponding to these functions.

## Figures and Tables

**Figure 1 jfb-15-00215-f001:**
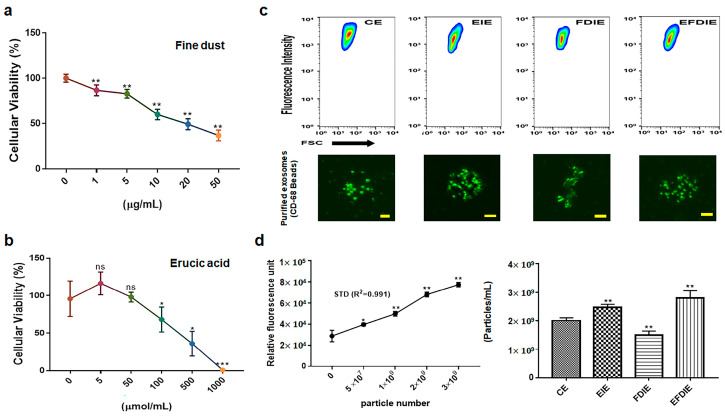
Establishment of treatment dosages for fine dust, erucic acid, and induction of exosomes in fibroblasts. Establishing treatment doses of fine dust and erucic acid for fibroblasts (**a**,**b**). Purification of exosomes isolated from fibroblasts under various conditions (CE, EIE, FDIE, EFDIE) and the exosomal images (**c**). The results indicate the concentration of exosomes (left; standard curve, right; bar graphs for their concentrations) isolated from fibroblasts under the various conditions (**d**). CE, control-induced exosomes; E, erucic acid-induced exosomes; FDIE, fine dust-induced exosomes; EIE, erucic acid; EFDIE; induced exosomes from fine dust after exposure to erucic acid; ns, not significant (* *p* < 0.05; ** *p* < 0.01; *** *p* < 0.001) (scale bars = 20 μm).

**Figure 2 jfb-15-00215-f002:**
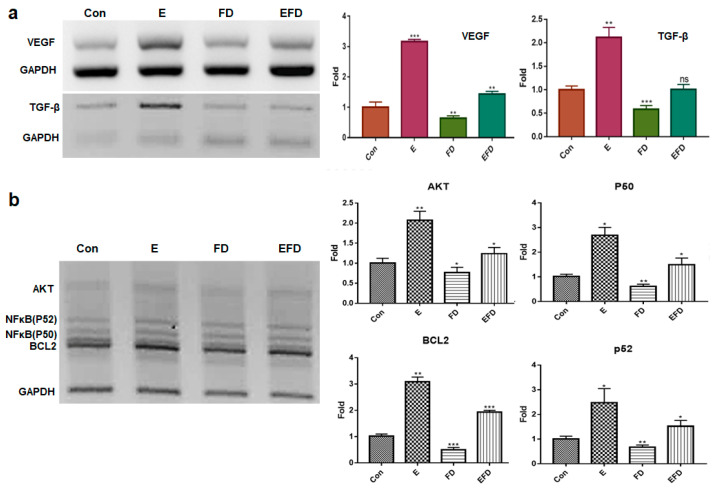
The levels of osteogenic activators and anti-apoptotic markers in fibroblasts under various exposure conditions. Levels of osteogenic activators VEGF and TGF-β in fibroblasts under different conditions (control, E, FD, and EFD) (**a**). Levels of anti-apoptotic markers in fibroblasts under different conditions (**b**). Con, control; FD, fine dust; E, erucic acid; EFD, exposure to FD after E exposure) ns, not significant; (* *p* < 0.05; ** *p* < 0.01; *** *p* < 0.001).

**Figure 3 jfb-15-00215-f003:**
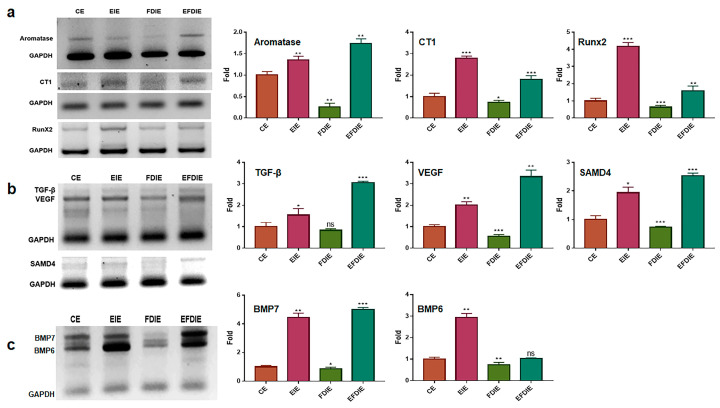
The levels of osteocytic differentiation markers in adipose-derived stem cells in the induced exosomes isolated from various conditions. The levels of markers for osteocytic differentiation in adipose-derived stem cells under various conditions (CE, EIE, FDIE, EFDIE) (**a**–**c**). The induced exosomes were isolated from fibroblasts under four conditions. CE, control-induced exosomes; FDIE, fine dust-induced exosomes; EIE, erucic acid-induced exosomes; EFDIE, FD-induced exosomes after exposure, ns, not significant; (* *p* < 0.05; ** *p* < 0.01; *** *p* < 0.001).

**Figure 4 jfb-15-00215-f004:**
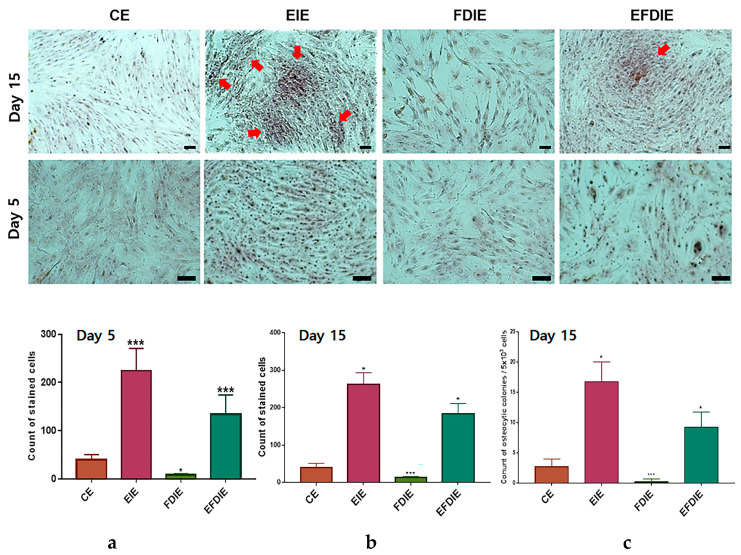
Osteocytic differentiation in adipose-derived stem cells with the induced exosomes isolated from various conditions. Alizarin staining for osteocytic differentiation in adipose-derived stem cells under various conditions (CE, EIE, FDIE, EFDIE) (**a**–**c**). The red arrows indicate osteocytic colonies, and the black spots in the cells in the stained images are calcium granules. The induced exosomes were isolated from fibroblasts under four conditions: CE, control-induced exosomes; FDIE, fine dust-induced exosomes; EIE, erucic acid-induced exosomes; EFDIE, FD-induced exosomes after erucic acid exposure; (* *p* < 0.05; *** *p* < 0.001), (the scale bars = 30 μm).

**Figure 5 jfb-15-00215-f005:**
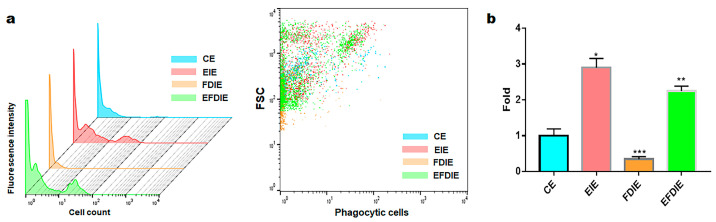
Phagocytic activity of macrophages with induced exosomes isolated from various conditions. Stagger-offset (**Left**) and dot plot for the phagocytosis-positive cells with the induced exosomes (**right**) (**a**) and the results of counting the positive cells compared with the control (**b**). CE, control-induced exosomes; FDIE, fine dust-induced exosomes; EIE, erucic acid-induced exosomes; EFDIE, FD-induced exosomes after E exposure (* *p* < 0.05; ** *p* < 0.01; *** *p* < 0.001).

**Figure 6 jfb-15-00215-f006:**
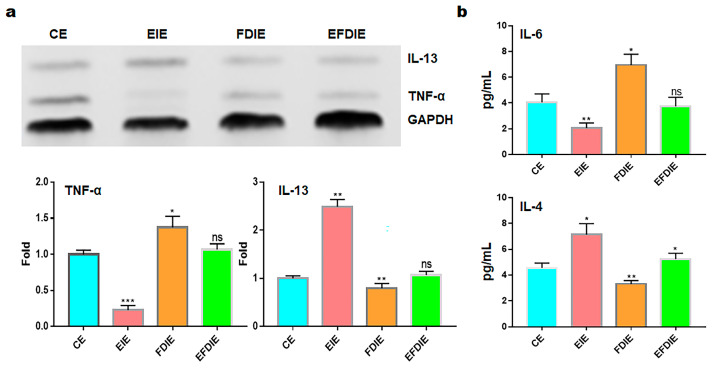
The levels of the osteogenesis-modulating cytokines in macrophages with induced exosomes. Panels (**a**,**b**) show the PCR and ELISA data, respectively. Cytokine levels in macrophages (**a**) and secreted cytokines in the supernatants of macrophages under various conditions. CE, control-induced exosomes; FDIE, fine dust-induced exosomes; EIE, erucic acid-induced exosomes; EFDIE, FD-induced exosomes after exposure; ns, not significant; (* *p* < 0.05; ** *p* < 0.01; *** *p* < 0.001).

**Figure 7 jfb-15-00215-f007:**
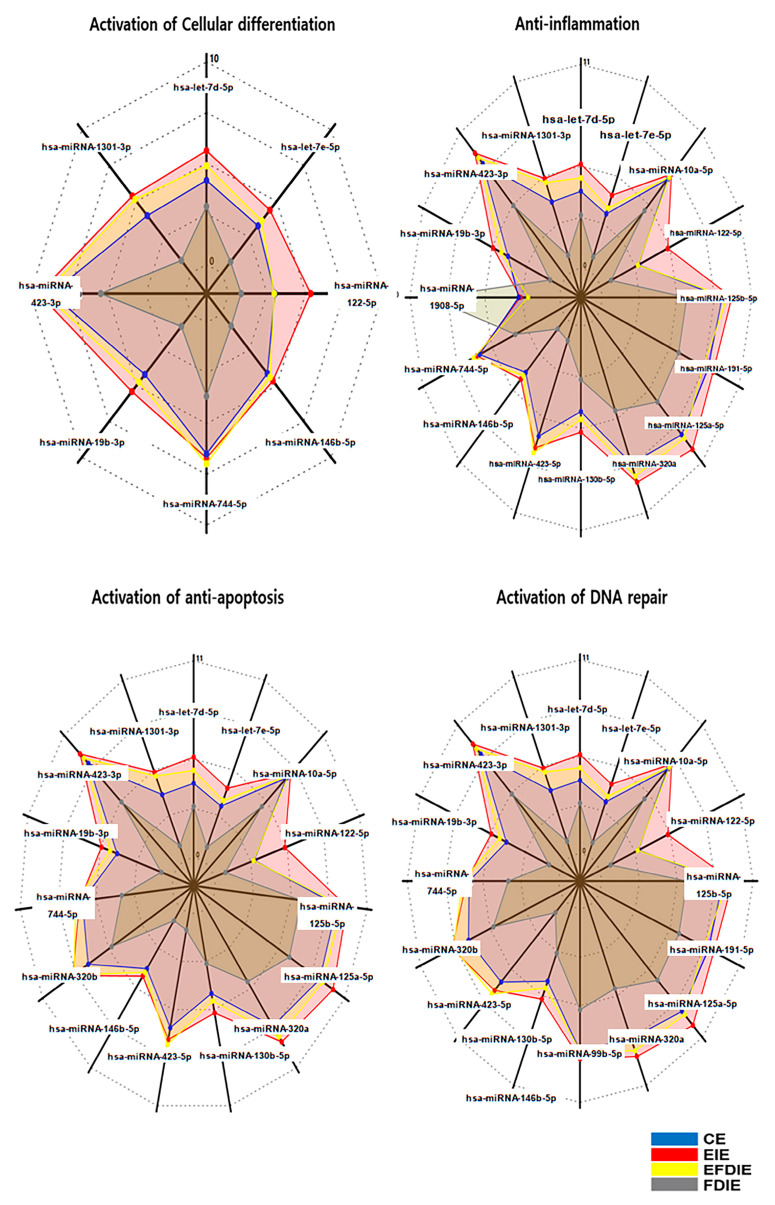
Profiling of miRNAs in various exosomes. The radar charts display significantly altered miRNAs in the various exosomes derived from gingival cells. Red lines: EIE, yellow: EFDIE, blue lines: CE and gray lines: FDIE, (*p* < 0.05).

**Table 1 jfb-15-00215-t001:** The list of primers for qRT-PCR.

Gene	Band Size (bp)	F/R*	Seq (5’ → 3’)
*AKT*	682	F	GGCTGCCAAGTGTCAAATCC
		R	AGTGCTCCCCCACTTACTTG
*NFκB-P50*	480	F	CGGAGCCCTCTTTCACAGTT
		R	TTCAGCTTAGGAGCGAAGGC
*NFκB-P52*	410	F	AGGTGCTGTAGCGGGATTTC
		R	AGAGGCACTGTATAGGGCAG
*Bcl2*	386	F	CTGCTGACATGCTTGGAAAA
		R	ATTGGGCTACCCCAGCAATG
*IL13*	361	F	CCACGGTCATTGCTCTCACT
		R	CCCGCCTACCCAAGACATTT
*TNF-α*	262	F	AAGAGGGAGAGAAGCAACTAC
		R	AGGAGAAGAGGCTGAGGAAC
*Aromatase*	287	F	TCAGAGCAACCTTCTTAGGCTC
		R	AGAAAAGTTACCTGAGAGGCCA
*CT1*	542	F	GCTCGTGGAAATGATGGTGC
		R	CCTCGCTTTCCTTCCTCTCC
*RunX2*	470	F	TTGCAGCCATAAGAGGGTAG
		R	GTCACTTTCTTGGAGCAGGA
*BMP7*	577	F	CCCGGGTAGCGCGTAG
		R	CGTTCCCGGATGTAGTCCTT
*BMP6*	520	F	CTTCCCATCCTTTCTGCGAGC
		R	GGGCCACCATGAAGTTTACC
*TGF-β*	369	F	CTGTCCAACATGATCGTGCG
		R	AGTGCCCAAGGTGCTCAATA
*VEGF*	316	F	ACTGCCATCCAATCGAGACC
		R	GAAGGCAAGACCCCACCATA
*SAMD4*	247	F	TCTTGACAGTGTTCCACGGG
		R	GCAAAGCCAAGGAAGCACAT
*GAPDH*	210	F	GTGGTCTCCTCTGACTTCAACA
		R	CTCTTCCTCTTGTGCTCTTGCT

(bp; base pair).

**Table 2 jfb-15-00215-t002:** Profiling of dramatic changed miRNAs in erucic acid-induced exosomes.

Categories	Activation of Differentiation	Anti-Inflammation	Anti-Apoptosis	Activation of DNA Repair
Upregulation Genes	hsa-let-7d-5p	hsa-let-7d-5p	hsa-let-7d-5p	hsa-let-7d-5p
	hsa-let-7e-5p	hsa-let-7e-5p	hsa-let-7e-5p	hsa-let-7e-5p
	hsa-miRNA-1301-3p *	hsa-miRNA-1301-3p	hsa-miRNA-1301-3p	hsa-miRNA-1301-3p
	hsa-miRNA-423-3p	hsa-miRNA-423-3p	hsa-miRNA-423-3p	hsa-miRNA-423-3p
	hsa-miRNA-19b-3p	hsa-miRNA-19b-3p	hsa-miRNA-19b-3p	hsa-miRNA-19b-3p
	-	hsa-miRNA-1908-5p *	-	-
	hsa-miRNA-744-5p	hsa-miRNA-744-5p	hsa-miRNA-744-5p	hsa-miRNA-744-5p
	hsa-miRNA-146b-5p	hsa-miRNA-146b-5p	hsa-miRNA-146b-5p	-
	-	hsa-miRNA-423-5p *	hsa-miRNA-423-5p *	hsa-miRNA-423-5p *
	-	hsa-miRNA-130b-5p	hsa-miRNA-130b-5p	hsa-miRNA-130b-5p
	-	hsa-miRNA-320a	hsa-miRNA-320a	hsa-miRNA-320a
	-	-	hsa-miRNA-320b	hsa-miRNA-320b
	-	hsa-miRNA-125a-5p	-	hsa-miRNA-125a-5p
	-	hsa-miRNA-191-5p	-	hsa-miRNA-191-5p
	-	hsa-miRNA-125b-5p	hsa-miRNA-125b-5p	hsa-miRNA-125b-5p
	hsa-miRNA-122-5p	hsa-miRNA-122-5p	hsa-miRNA-122-5p *	hsa-miRNA-122-5p
	-	hsa-miRNA-10a-5p	hsa-miRNA-10a-5p	hsa-miRNA-10a-5p
	-	-		hsa-miRNA-99b-5p

*: 4 Folds.

## Data Availability

The original contributions presented in the study are included in the article, further inquiries can be directed to the corresponding author.

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
