# Peer review of "Osteogenic Protection against Fine Dust with Erucic Acid-Induced Exosomes"

_jfb, 2024, doi:10.3390/jfb15080215_

Round 1

Reviewer 1 Report

Comments and Suggestions for Authors

Review report for Manuscript ID: jfb-3099293

The manuscript titled “Osteogenic Protection of Erucic Acid-Induced Exosomes Against Fine Dust,” authored by Hyunjung Kim and Boyong Kim, presents a comprehensive study on the effects of erucic acid-induced exosomes on osteogenesis and inflammatory responses. The experimental design was meticulous, and the findings are significant. This study makes a valuable contribution to the efforts against the adverse effects of fine dust.  I think it is suitable to be published in JFB if authors address the following comments.

Specific comments are:

1.        The paragraph including lines 63-73 lists the factors involved in inflammation and bone formation but lacks organization. If there is an interplay between these two processes, please describe it. Otherwise, separate them into two different paragraphs. In this aspect, the Discussion section was more organized and streamlined.

2.        The authors should describe the detrimental effects of fine dust on bone diseases and the current treatments or research on these treatments. It would be helpful for the authors to present the questions they aim to address and introduce their strategies in the introduction. 

3.        Why was the erucic acid concentration selected at 250 µmol/mL? Figure 1b shows that 250 µmol induced about 50% cell death. Is the 250 µmol in Figure 1b equivalent to 250 µmol/mL? Clarification is needed.

4.        The authors should include images of fibroblasts with and without treatment to show the cell morphologies. These images may be included in the Supplemental Materials.

5.        In the figure legends (e.g., Figures 3 and 4), the authors should indicate the medium in which ASCs were cultured and the concentrations of exosomes added. A positive control of osteogenic differentiation of ASCs induced by conventional osteogenic induction medium (e.g., dexamethasone) would be important to gauge the activity of ELE.

6.        Does erucic acid directly enhance the osteogenic differentiation of ASCs? The authors should demonstrate the advantage of using EIE over erucic acid.

7.        Is EVERBIO a business or corporation? If so, the authors should indicate any potential conflicts of interest.

Other comments:

Line 55: “For this high exposure rate of PM10, PM10 affects a worse condition for a skin” needs to be clarified.

Line 109: What does “respectively” mean? Four groups of samples used two different methods separately.  Or two methods were used for all four samples?

Comments on the Quality of English Language

Author Response

Comments

  1. The paragraph including lines 63-73 lists the factors involved in inflammation and bone formation but lacks organization. If there is an interplay between these two processes, please describe it. Otherwise, separate them into two different paragraphs. In this aspect, the Discussion section was more organized and streamlined.
  2. The authors should describe the detrimental effects of fine dust on bone diseases and the current treatments or research on these treatments. It would be helpful for the authors to present the questions they aim to address and introduce their strategies in the introduction. 
  3. Why was the erucic acid concentration selected at 250 µmol/mL? Figure 1b shows that 250 µmol induced about 50% cell death. Is the 250 µmol in Figure 1b equivalent to 250 µmol/mL? Clarification is needed.
  4. The authors should include images of fibroblasts with and without treatment to show the cell morphologies. These images may be included in the Supplemental Materials.
  5. In the figure legends (e.g., Figures 3 and 4), the authors should indicate the medium in which ASCs were cultured and the concentrations of exosomes added. A positive control of osteogenic differentiation of ASCs induced by conventional osteogenic induction medium (e.g., dexamethasone) would be important to gauge the activity of ELE.
  6. Does erucic acid directly enhance the osteogenic differentiation of ASCs? The authors should demonstrate the advantage of using EIE over erucic acid.
  7. Is EVERBIO a business or corporation? If so, the authors should indicate any potential conflicts of interest.    

    Answers

    First, we appreciate for your comments to improve out manuscript.

    We revised our manuscript based on your comments and marked the revised sentences with underlines, a red color. The revised sentences based on common comments from reviewers were underlined in yellow color. 

    1] We added description and reference for the interplay in the paragraph.

    2] We already described the relationship between fine dust and bone diseases at line 54 to 62 but, we described goal of this study more detail in the introduction section at line 77 to 82

    3] We revised the panel b in fig 1.

    4] We added the Images of Fibroblasts under various conditions in supplementary data      

    5] Your comment is very helpful, and we have discussed whether to use positive control like that compound before this study. However, our goal is to develop a liposomal material with microRNAs. We would like to apply your helpful idea in further studies with synthesized liposomes.

    6] In our screening experiments, EIE (Erucic Acid-Induced Exosomes) was more effective than erucic acid in promoting osteogenic differentiation. We have included these results in the supplementary data.

    7] We already added “Conflicts of Interest: The authors declare no conflict of interest” in the manuscript

Reviewer 2 Report

Comments and Suggestions for Authors

The manuscript written by Hyunjung Kim et al. entitled “Osteogenic protection of erucic acid-induced exosomes against fine dust” looks into the beneficial effects of erucic acid upon exposure of fibroblast to fine dust and the effects of their exosomes on adipose mesenchymal stromal cells osteogenic differentiation and macrophages inflammatory profile. Altought the work is interesting, some issues of utter importance must be addressed before considering the work for publication.

Comments:

Introduction:

The ending paragraph is vague, unclear and lacks connection with the rest of the introduction. Please rewrite it.

Materials and methods:

In line 138 the standard says 44. I guess it should say 4.4

In section 2.9, please add the p values considered statistically significant.

Results:

Why did the authors chose the concentrations of fine dust and erucic acid in which the cellular viability is of about 50% to perform the experiments? These conditions are not optimal for the evaluation of the extracellular vesicles, as cells should be vital. Authors must justify these choice well, as this is a critical point of the work.

In Figure 1, panel C is not legible and does not really provide any information. Same for panel D. The standard curve could be included as a supplementary file or omitted at all. Authors should provide a table or a chart with the protein concentration in the samples, not the absorbance.

The extracellular vesicles must be characterized according to the international society for extracellular vesicles (ISEV) (Thery 2018). A new figure with the whole characterization must be included, as this is critical for the article. Specially, due to the possible contamination with apoptotic bodies, given the cellular viability of the fibroblasts. Also, according to the ISEV, authors should demonstrate the endosomal origin of extracellular vesicles, in order to call them exosomes. If not, the term “extracellular vesicles” is preferred.

In figure 2, some panels look stretched; please improve the quality of the images. The same for figure 3. Authors may consider showing data as a heat map, in order to simplify the data and make its understanding more straightforward. In panel C, why does the GAPDH looks so faint?

Figure 4: Please make the graphics consistent throughout the manuscript, keeping the same colors/bar formats for the same conditions. What about the quantifications at day 5?

Figure 5 is utterly redundant. Panel A, B, and C provide the same information. Figure 5 and 6 should be unified, as they show the effect of the EVs on macrophages.

In table 2, avoid the use of “dramatic changes” for variations in the levels of mRNA. Please use an appropriate expression.

Figure 7 is not clear and is not legible. Please improve its quality and include the reference of which color corresponds to each treatment.

It is known that fine dust induces oxidative stress, Why did the authors not evaluate the effect of the extracellular vesicles on oxidative stress? It would add great value to the manuscript

Discussion:

I would suggest discussion the results in the order of appearance in the manuscript. The way the authors discuss the results ( First figure 2, then figure 7 and then figure 2 again, etc), is confusing and not clear.

Authors claim that “First, erucic acid prevented fibroblasts from being inflamed upon exposure to 20 ng/mL fine dust (Figure 2)”. However, data do not support this conclusion. Authors only show that NFkB expression is diminished.

Authors consider EIE as “biomaterials”, but I think this term is misleading.

The conclusion is not related to the content of the manuscript.  Please rewrite it.

Comments on the Quality of English Language

Minor editing required

Author Response

Comments

Introduction:

The ending paragraph is vague, unclear and lacks connection with the rest of the introduction. Please rewrite it.

Materials and methods:

In line 138 the standard says 44. I guess it should say 4.4

In section 2.9, please add the p values considered statistically significant.

Results:

Why did the authors chose the concentrations of fine dust and erucic acid in which the cellular viability is of about 50% to perform the experiments? These conditions are not optimal for the evaluation of the extracellular vesicles, as cells should be vital. Authors must justify these choice well, as this is a critical point of the work.

In Figure 1, panel C is not legible and does not really provide any information. Same for panel D. The standard curve could be included as a supplementary file or omitted at all. Authors should provide a table or a chart with the protein concentration in the samples, not the absorbance.

The extracellular vesicles must be characterized according to the international society for extracellular vesicles (ISEV) (Thery 2018). A new figure with the whole characterization must be included, as this is critical for the article. Specially, due to the possible contamination with apoptotic bodies, given the cellular viability of the fibroblasts. Also, according to the ISEV, authors should demonstrate the endosomal origin of extracellular vesicles, in order to call them exosomes. If not, the term “extracellular vesicles” is preferred.

In figure 2, some panels look stretched; please improve the quality of the images. The same for figure 3. Authors may consider showing data as a heat map, in order to simplify the data and make its understanding more straightforward. In panel C, why does the GAPDH looks so faint?

Figure 4: Please make the graphics consistent throughout the manuscript, keeping the same colors/bar formats for the same conditions. What about the quantifications at day 5?

Figure 5 is utterly redundant. Panel A, B, and C provide the same information. Figure 5 and 6 should be unified, as they show the effect of the EVs on macrophages.

In table 2, avoid the use of “dramatic changes” for variations in the levels of mRNA. Please use an appropriate expression.

Figure 7 is not clear and is not legible. Please improve its quality and include the reference of which color corresponds to each treatment.

It is known that fine dust induces oxidative stress, Why did the authors not evaluate the effect of the extracellular vesicles on oxidative stress? It would add great value to the manuscript

Discussion:

I would suggest discussion the results in the order of appearance in the manuscript. The way the authors discuss the results (First figure 2, then figure 7 and then figure 2 again, etc), is confusing and not clear.

Authors claim that “First, erucic acid prevented fibroblasts from being inflamed upon exposure to 20 ng/mL fine dust (Figure 2)”. However, data do not support this conclusion. Authors only show that NFkB expression is diminished.

Authors consider EIE as “biomaterials”, but I think this term is misleading.

The conclusion is not related to the content of the manuscript.  Please rewrite it.

Answers

  1. Introduction

We revised more detailly the end paragraph in the introduction

  1. Materials and methods:

We revised the line 138 and added the sentence for the p values considered statistically significant in the section 2.9

  1. Results

3.1] Typically, CC50 is used to determine the concentration of a substance required to induce stimulation in cells and has been frequently employed in my studies on natural products and exosomal microRNA change analyses, published in SCI(E) journals. Through this research, foundational data on the functionality of exosomes and optimal treatment concentrations are provided. The altered microRNA profiling induced within exosomes aims to offer crucial data for developing liposomal pharmaceutical materials.

3.2] We entirely revised fig1 based on your comments

3.3] We cultured the cells without FBS to avoid contamination from other extracellular vesicles. We isolated only cell-derived exosomes. To avoid confusion, we added this description in Section 2.3 of the Materials and Methods.

3.4] We revised resolution and quality of the fig 2 and 3.  

3.5] We revised graph types in the fig 2.

3.6] We revised the fig 4 based on your comment

3.7] We revised the fig5. In fig5 and 6, TNF-α inhibits the clearance of apoptotic cells by macrophages, while IL-10 and IL-4 activate phagocytosis by macrophages. In the results shown in Figure 6, EIE significantly upregulated IL-13 and IL-4, which are anti-inflammatory cytokines in macrophages. Despite TNF-α typically activating macrophage phagocytosis, EIE attenuated these levels in macrophages. These results suggest that the elevated levels of IL-13 and IL-4 played a crucial role in activating phagocytosis in macrophages (Figure 5 and 6). We added the

3.8] We revised the table 2

3.9] We revised the fig7

3.10] Ordinarily, oxidative stress induces inflammation and apoptosis in cells. Furthermore, inflammation and apoptosis inhibit osteogenesis from stem cells. In our manuscript, we described the functions of EIE, including its anti-apoptotic effects and the upregulation of anti-inflammatory cytokines. For these reasons, we excluded these evaluations in this study.

  1. Discussion:

4-1] We rearranged the discussion more clearly

4-2] We revised the conclusion section based on your comment.  

4-3] we revised the sentence

4-4] There are two common definitions of biomaterials. First, a material derived from, or produced by, biological organisms like plants, animals, bacteria, fungi and other life forms. These are also called biologically derived materials. Second, a material used for a biological purpose such as a biomedical application like treating an injury or growing biological cells. In my opinion, due to these definitions and, description about exosomes of the introduction section, there will be few confusing for the word.

Round 2

Reviewer 1 Report

Comments and Suggestions for Authors

 I am satisfied with the revision of the manuscript and consider it is acceptable.

Author Response

Thank you very much for your positive decision 

Reviewer 2 Report

Comments and Suggestions for Authors

I thank the authors for assessing some my concerns. I believe that the quality of the manuscript has substantially improved. Still, I consider that some points should be addressed:

The main flaw of the manuscript is that authors affirm that extracellular vesicles mediate the anti-inflammatory and anti-apoptotic effect observed. However, authors do not demonstrate the presence of EVs. Authors must provide, at least, a size and distribution profile, transmission/scanning electron microscopy images, presence of positive exosomes markers (CD9,CD63, CD81, Alix, TSG101, etc) and abscense of negative markers (calnexin, cytochrome C, etc), according to the ISEV. Without this characterization, authors cannot affirm that there are extracellular vesicles in their preparations and cannot exclude the presence of other vesicles such as microvesicles, ectosomes and/or apoptotic bodies.

Also, in figure 1D, it is not clear why authors show results in ug/mL if the quantification kit provides the concentration in particles/mL. Did they quantify the protein concentration in the samples? Please clarify.

Authors did not respond why in some blots GAPDH is so faint (more than the proteins of interest) and in others the image is almost saturated. Authors must provide all the molecular weights of the studied proteins in the figures. Moreover, the full blot images provided MUST include the molecular weight marker in the blot, to allow the direct observation. Without this, the veracity of the blots is questioned.

Aging, the conclusion is pretentious and is not supported by the results. Authors claim that “In this manuscript, we demonstrate a new biomaterials, EIE can prevent bone diseases caused by fine dust, in addition to its immune modulation and anti-inflammatory properties.” Authors only show the in vitro effect of the EIE, and do not provide any evidence in animal models showing that EIE can prevent bone diseases. Please provide a conclusion supported by the results presented in the manuscript.  English editing is required.

Comments on the Quality of English Language

English editing is required.

Author Response

Thank you for your comments, our manuscript is more improved by your help. 

Comments 1] The main flaw of the manuscript is that authors affirm that extracellular vesicles mediate the anti-inflammatory and anti-apoptotic effect observed. However, authors do not demonstrate the presence of EVs. Authors must provide, at least, a size and distribution profile, transmission/scanning electron microscopy images, presence of positive exosomes markers (CD9,CD63, CD81, Alix, TSG101, etc) and abscense of negative markers (calnexin, cytochrome C, etc), according to the ISEV. Without this characterization, authors cannot affirm that there are extracellular vesicles in their preparations and cannot exclude the presence of other vesicles such as microvesicles, ectosomes and/or apoptotic bodies.

Answer1] We added the images of purified exosomes at fig1c

Comments 2] Also, in figure 1D, it is not clear why authors show results in ug/mL if the quantification kit provides the concentration in particles/mL. Did they quantify the protein concentration in the samples? Please clarify.

Answer2] First, I apology the mistake. I misunderstood the provided data for the std curves from first author, Hyunjung Kim. I revised fig1 and result section clearly. 

Comments 3] Authors did not respond why in some blots GAPDH is so faint (more than the proteins of interest) and in others the image is almost saturated. Authors must provide all the molecular weights of the studied proteins in the figures. Moreover, the full blot images provided MUST include the molecular weight marker in the blot, to allow the direct observation. Without this, the veracity of the blots is questioned.

Answer3] We change gel-doc with clearer GAPDH

Aging, the conclusion is pretentious and is not supported by the results. Authors claim that “In this manuscript, we demonstrate a new biomaterials, EIE can prevent bone diseases caused by fine dust, in addition to its immune modulation and anti-inflammatory properties.” Authors only show the in vitro effect of the EIE, and do not provide any evidence in animal models showing that EIE can prevent bone diseases. Please provide a conclusion supported by the results presented in the manuscript.  English editing is required.

Answer4] We revised the conclusion section more clearly and got English edition again 

Round 3

Reviewer 2 Report

Comments and Suggestions for Authors

Authors did not addressed all of my concerns. Still,  the presence of the EVs is not demostrated and they need to be characterized according to the ISEV. Therefore, authors cannot claim that the observed effect is mediated by EVs. Authors must provide this characterization, or otherwise consider rewriting the manuscript and consider that the observed effects are mediated by "soluble factors". In the present form, the manuscript is not suitable for publication, as results do not support the conclusions.

Also, authors did not provide full blots with the corresponding molecular weight markers, which I had requested.

Comments on the Quality of English Language

Extensive editing of English language required

Author Response

Comment] Authors did not addressed all of my concerns. Still,  the presence of the EVs is not demostrated and they need to be characterized according to the ISEV. Therefore, authors cannot claim that the observed effect is mediated by EVs. Authors must provide this characterization, or otherwise consider rewriting the manuscript and consider that the observed effects are mediated by "soluble factors". In the present form, the manuscript is not suitable for publication, as results do not support the conclusions.

Also, authors did not provide full blots with the corresponding molecular weight markers, which I had requested.

Answer] To isolate the induced exosomes, we cultured cells without FBS and applied the exosome isolating kits (the exoEasy Maxi Kit for Exosome Isolation; Qiagen: The kit isolates only extracellular vesicles ranging from 50 to 300 nm.) and the purifying kit (anti CD-68 coated beads) to the supernatants from stimulated cells. The data from two kits and flow cytometry can replace the western blot marker data. Additionally, we already displayed the band size markers in the qPCR primer table and full gels (supplementary data) and provided only qPCR data in this manuscript. To clarify this question, we have added flow cytometry data using FITC-CD-68 beads in the supplementary data. In my numerous published papers in SICE Q1 group journals, exosomes purified using the two kits have been recognized as functional biomaterials.